# HOLMES & WATSON: A Robust and Lightweight HTTPS Website Fingerprinting through HTTP Version Parallelism

## Abstract

Website Fingerprinting (WF) is a traffic analysis technique that aims to identify websites visited by users through the analysis of encrypted traffic patterns. Existing approaches often exhibit limited robustness against network variability and concept drift, resulting in significant performance degradation under real-world HTTPS conditions. Moreover, these methods typically require large-scale training datasets and substantial computational resources, which further increases the complexity of deployment. In this paper, we propose HOLMES, a novel approach that exploits HTTP version parallelism to extract enhanced application-layer features. These features, including the number of web resources transmitting in various HTTP versions, expose up to 4.28 bits of information—surpassing 98% of previously reported features and demonstrate increased stability across varying network conditions. Complementary to this, we introduce WATSON, a lightweight classification method based on lazy learning, which substantially reduces the dependency on large training datasets. To further enhance the identification accuracy, we incorporate two fingerprint-specific distance metrics that ensure high intra-class similarity. Our experimental evaluation demonstrates that HOLMES & WATSON significantly enhance both robustness and efficiency, achieving an average accuracy of 87.7% with only a single sample per website, marking an improvement of over 15% compared to state-of-the-art methods.

## CCS Concepts

• **Networks → Network privacy and anonymity**; **Application layer protocols**; • **Security and privacy → Pseudonymity, anonymity and untraceability**.

## Keywords

Website fingerprinting, HTTP version parallelism, Protocol analysis, Lazy learning

**ACM Reference Format:**

## 1 INTRODUCTION

The increasing adoption of HTTPS has significantly improved user privacy by encrypting communication content. However, adversaries can still deduce web activities by exploiting side-channel information, such as DNS queries [32], the TLS Server Name Indication (SNI) field [89], and corresponding server IP addresses [53]. To mitigate these privacy vulnerabilities, privacy-enhancing technologies such as encrypted DNS (DoH [34], DoT [23], DoQ [36]) and TLS Encrypted Client Hello (ECH [17, 62]) have been introduced. Additionally, the increasing prevalence of IP co-location,

*WWW'25, April 2025, Sydney, Australia*
© 2024 Copyright held by the owner/author(s). Publication rights licensed to ACM.
ACM ISBN 978-x-xxxx-xxxx-x/YY/MM
https://doi.org/10.1145/nnnnnnn.nnnnnnn

**Figure 1: Five main kinds of methods to identify websites from encrypted traffic: (a) Using IP addresses with entropy-based matching. (b) Using traffic trace sequence features with deep learning. (c) Using traffic statistical features with machine learning. (d) Using TLS SNI parameters. (e) Using web resource features with lazy learning.**

largely facilitated by content delivery networks (CDNs), has further diminished the effectiveness of IP-based identification methods[6].

While these countermeasures reduce the efficacy of traditional side-channel attacks, they do not fully eliminate the risk posed by traffic analysis. Notably, website fingerprinting (WF) attacks continue to represent a significant threat [5, 21, 29, 31, 65, 70, 87]. WF enables adversaries to analyze patterns in encrypted traffic and infer the websites a user visits [67]. Although the majority of research on WF has been concentrated on anonymity networks, such as Tor [24], there is growing interest in applying these techniques to HTTPS due to their potential impact on encrypted web traffic.

However, existing WF attacks struggle in realistic HTTPS scenarios due to their reliance on *unrealistic assumptions* [15, 39, 81] that conflict with the actual nature of HTTPS traffic. These assumptions often include stable network conditions, the absence of concept drift, and the availability of large labeled datasets, all of which are misaligned with the dynamic and diverse nature of network traffic. While several work have demonstrated high accuracy in controlled environments [7, 29, 52, 58, 70, 71], their performance degrades substantially in realistic HTTPS scenarios. For instance, Juarez et al. [15] showed that concept drift leads to a significant reduction in WF accuracy, and our experiments corroborate this finding, revealing a 20% drop in accuracy under varying network conditions (§4.4). Furthermore, the training process for a typical WF classifier requires over 10 days of data collection [71], which presents significant limitations for its practical deployment.

We argue that the limitations mentioned above largely stem from inadequate feature extraction layer. Existing WF attacks mainly rely on transport-layer features, such as packet direction sequences and inter-packet delays. As shown in Figure 1, these features, situated at

the transport layer of the TCP/IP model, are distant from the actual characteristics of website resources, making them more vulnerable to network condition changes and less robust [85]. Moreover, these transport-layer features are often sparse and fragmented, requiring complex machine learning or deep learning models, as well as large training datasets, which limits their practicality [13].

To address these challenges, we propose a novel WF method that leverages features at the application layer. Despite content encryption and multiplexing in HTTPS, signals such as resource count and connection numbers can be inferred from the browser request side [3, 48]. More importantly, we identified a growing phenomenon in the modern internet ecosystem: **HTTP version parallelism**. Due to multiple HTTP versions running concurrently without mandatory upgrades, website owners independently choose which version to adopt. This results in distinct patterns across HTTP versions for different websites, as illustrated in Figure 2(a). By expanding the feature space with HTTP version parallelism, we significantly enhance the distinguishing power of application-layer features; for instance, the quantity of HTTP/2 resources alone reveals 4.28 bits of information (see Figure 2(b)), exceeding 98% of existing features. Moreover, these features demonstrate greater stability across varying network conditions, as shown in Figure 2(c).

Based on the identified features, we propose a rich and robust website fingerprint representation—H123 fingerprint—utilizing characteristics such as web resource quantity, HTTP versions, and loading sequences. To differentiate websites using this representation, we employ a hybrid distance metric that combines Wasserstein distance and a modified Longest Common Subsequence (LCS) metric, assigning different weights to enhance their contributions to fingerprint similarity assessment. Considering practical requirements for a lightweight solution, we adopt a **Lazy Learning** paradigm (e.g., k-Nearest Neighbor [18]) that computes predictions on demand, eliminating the need for prior model training [2]. This approach allows for rapid deployment and flexible adjustments to the monitored set of websites. Moreover, by relying on direct distance measurements rather than model fitting, our method requires only a minimal number of reference samples for effective WF attack. Notably, our approach attains an average accuracy of 87.7% with only one sample per website, reflecting an enhancement of more than 15% compared to current state-of-the-art methods.

Our contributions are as follows:

- We are the first to identify and analyze the privacy risks posed by application-layer features under the context of modern HTTP version parallelism. We propose the **H**TTP res**O**urce extrapo**L**ating **M**ethod under **E**ncrypted **S**cheme (**HOLMES**), which extracts these features without decryption, enabling the generation of robust and information-dense H123 fingerprints.
- We present the **WA**sserstein and **T**extual **S**imilarity-based rec**O**g**N**izer (**WATSON**), a lightweight website fingerprinting attack method based on lazy learning and specific distance measures that performs well even in few-shot scenarios.
- We collect and publish[1] a comprehensive dataset containing over 1.5TB of HTTPS traffic, covering 220,000 samples from 80,000 websites across 12 experimental scenarios. This is the first HTTPS WF dataset to include traffic generated under challenging

---

[1]The dataset will be released before publishing.

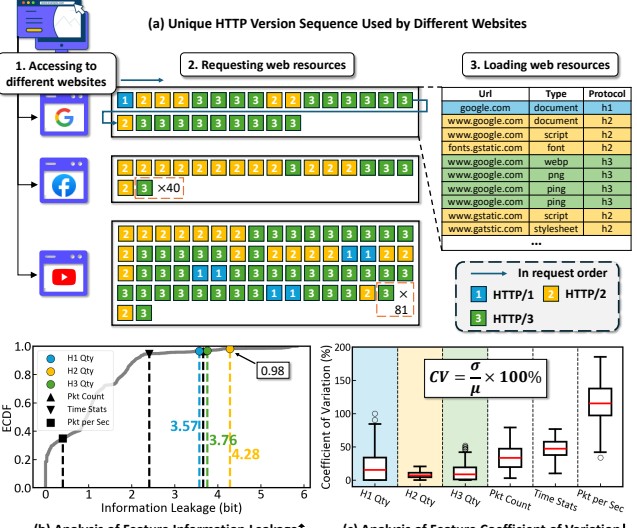

**Figure 2: HTTP Version Parallelism and Privacy Implications.** (a) Demonstrate HTTP version parallelism in popular websites. (b) Assess the information leakage [45] potential caused by the quantity of HTTP resources in relation to a known 3043-dimensional feature set (300 monitored websites). (c) Use the coefficient of variation [60] to illustrate the intra-class stability of various features in cross-network conditions.

network conditions, from multiple mainstream browsers, and with more realistic monitored website sets.

- We conducted comprehensive experiments, comparing our method with state-of-the-art approaches, demonstrating its effectiveness, robustness, and lightweight nature. We release the source code of H&W [4].

## 2 BACKGROUND AND MOTIVATION

### 2.1 HTTP Version Parallelism

The HyperText Transfer Protocol (HTTP) is fundamental to web browsing, enabling the transfer of content between clients and servers. Over time, HTTP evolved from HTTP/1.1 [26], which introduced persistent connections and chunked transfer encoding, to HTTP/2 [76] and HTTP/3 [8], which improved performance. HTTP/2's multiplexing and header compression reduced latency, while HTTP/3's use of QUIC [38] enhanced security and reliability. These advancements support modern encrypted web traffic, forming the backbone of HTTPS [61], where HTTP works with TLS or QUIC to protect user data from eavesdropping.

The phenomenon of concurrent HTTP versions has been somewhat acknowledged by the scientific community, primarily not in the context of privacy concerns. Discussions on the IETF mailing list regarding HTTP/3 have highlighted the challenges for browser and server implementers in maintaining support for all HTTP versions [37]. Nonetheless, we argue that the **inevitability**, **ubiquity**, and **distinctiveness** of HTTP version parallelism position it as a critical factor in potential privacy leakage.

**Figure 3: The overview of HOLMES & WATSON.**

**Inevitability of HTTP Version Parallelism.** Despite the increasing adoption of HTTP/3 due to its efficiency and stability, many websites continue to support older versions for compatibility [78]. Furthermore, the specific connection establishment process of HTTP/3 complicates its direct adoption; in most cases, an initial connection is made using an older HTTP version to exchange necessary information, after which data transmission rights are negotiated in a competitive manner [8, 50].

**Ubiquity of HTTP Version Parallelism.** In our survey of the top 100,000 websites (see Appendix B), we found that over 80% utilize at least two HTTP versions, with nearly half employing all three—HTTP/1.1, HTTP/2, and HTTP/3—during a single visit. This widespread parallelism has become a standard characteristic of modern web traffic.

**Distinctiveness of HTTP Version Parallelism.** Figure 2 illustrates how diverse patterns of HTTP version usage can form unique signatures. By combining various HTTP versions and the number of resources, these patterns create privacy risks that can be exploited to identify websites. This underscores the necessity for further investigation into how HTTP version parallelism impacts privacy.

## 2.2 Threat Model

This study follows an attacker capability assumption similar to that in existing WF research [5, 11, 35, 70, 87]. We assume a local passive eavesdropper (e.g., an ISP) positioned along the communication path between the user's terminal and the website server. The eavesdropper can monitor but not alter, delay, drop, or decrypt traffic, aiming to identify whether the user is visiting a monitored website and, if so, which one.

We also account for realistic environmental constraints faced by attackers, such as monitoring multiple website sets instead of focusing only on popular sites [33], using fewer samples for WF attacks [51, 71, 87], and testing under varying network conditions [5, 65, 87] and concept drift [5, 21, 33, 68]. Variations in browser behavior [35] and potential defense mechanisms [66, 68] are also considered. To maintain comparability with previous work and concentrate on privacy breaches, our attacks focus on websites' homepages. Although this may not fully represent real-world user behavior [15], Mitseva et al.[49] recently proposed a generalizable framework that could be applied to extend existing methods and address these limitations in future research.

## 3 METHODOLOGY

Unlike existing WF attacks, our approach extracts features at the traffic flow level [54]. Specifically, we generate a unified set of fingerprint features for each flow identified by a five-tuple: <*source IP*, *source port*, *destination IP*, *destination port*, *protocol*>. This analysis at the flow level offers a balance between the efficiency of packet-level features and the information richness of session-level features.

Figure 3 outlines the core components of our approach. In the following sections, we describe HOLMES for feature extraction and WATSON for fingerprinting attacks, following the typical steps of a WF attack.

## 3.1 HOLMES

As a first step, we define the direction of packets sent by the client as *out* and those received by the client as *in*. In typical HTTPS scenarios, a network connection first undergoes a transport-layer handshake, followed by a security-layer handshake, before entering the application-layer request-response cycle. The encryption at the security layer introduces packet obfuscation, which complicates efforts to focus solely on the HTTP phase [44]. Based on our analysis of protocol specifications, we identify the start of the HTTP phase as the first 'application data' TLS packet in the *out* direction, or, for HTTP/3, the first short-header QUIC packet in the *out* direction.

Moreover, we observe that metadata within the traffic can reveal the HTTP version and even the number of resources requested. This is because content encryption does not alter the HTTP request-response pattern, which varies across HTTP versions. Multiplexing, which affects the merging of responses, does not obscure the number of resources, as it can still be inferred from the request-side signals. Figure 12 offers a visual explanation of these characteristics.

*3.1.1 Inferring HTTP Versions.* HTTP/3 uses the novel QUIC protocol as its transport layer, enabling a simple distinction between HTTP/3 and earlier versions by checking whether QUIC (UDP) is in use. HTTP/1 lacks multiplexing (though pipelining exists, it is not widely supported by browsers [16, 83]), meaning that HTTP/1 requests and responses occur sequentially. This pattern is reflected in the first two packets of the HTTP phase, where we always see an "out" request followed by an "in" response. In contrast, HTTP/2 sends a "preface" packet before the first request to establish multiplexing configurations, resulting in the first two packets both being "out" requests. Thus, by observing the direction of the first

two packets in the HTTP phase, we can easily distinguish between HTTP/1 and HTTP/2.

*3.1.2 Inferring Resource Quantity.* We propose using the number of "out" packets in the HTTP phase to estimate the number of web resources requested. Through further protocol analysis and empirical testing across multiple browsers, we identify two main sources of estimation bias: (1) Setup/control packet bias—in HTTP/2 and HTTP/3, additional setup packets and control frames are sent, leading to an overestimation of resource count. We address this by filtering out small packets that do not carry actual content. (2) HTTP request method bias—when requests are sent via POST, browsers may split the request header and body into multiple packets, even though only one resource is being requested. We disregard this bias, as it consistently occurs and does not significantly affect the inference process.

Based on these observations, we design protocol state machines that implement these rules to accurately infer resource quantities across different HTTP versions. Details of this state machine design are presented in Appendix C.

*3.1.3 Constructing H123 fingerprint.* The H123 fingerprint is constructed by encoding each connection (i.e., traffic flow) based on its HTTP version and corresponding resource quantity. For $x$ connections, let $\text{HTTP}_i \in \{1, 2, 3\}$ represent the HTTP version of the $i$-th connection, and $R_i$ denote the resource quantity. The fingerprint is represented by a matrix $F \in \mathbb{R}^{3 \times x}$, where each column corresponds to a connection, and the rows represent HTTP/1, HTTP/2, and HTTP/3.

The value $F_{v,i}$ at row $v$ and column $i$ is defined as:

$$F_{v,i} = \begin{cases} R_i, & \text{if } \text{HTTP}_i = v \\ 0, & \text{otherwise} \end{cases} \tag{1}$$

where $v \in \{1, 2, 3\}$. This matrix encodes the HTTP version and resource quantity for each connection, forming the H123 fingerprint. Additionally, it is interesting to observe how different websites perform under the proposed H123 fingerprint. To illustrate this more clearly, we provide examples of H123 fingerprints for some of the most popular websites in Appendix D.

## 3.2 WATSON

*3.2.1 Lazy Learning.* Lazy learning (also known as instance-based learning) defers the training process to the prediction stage, unlike eager learning models such as decision trees or neural networks, which build a model during training [2]. In lazy learning, the algorithm makes predictions by comparing a new sample directly with stored reference samples, computing similarity or distance metrics to assign the sample to a category.

In the context of WF attacks, lazy learning offers the advantage of not requiring model pre-training, thus allowing for flexible addition or updating of website categories and samples [71, 80]. Additionally, by computing distances at the prediction stage, the method avoids overfitting risks commonly associated with insufficient training data.

To improve the efficiency of lazy learning, especially with large datasets, we introduced a resource-based sample pre-selection process. During prediction, a band-pass filter is applied to the reference

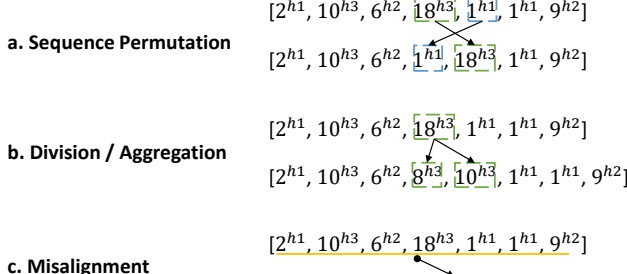

**a. Sequence Permutation**

$$[2^{h1}, 10^{h3}, 6^{h2}, 18^{h3}, 1^{h1}, 1^{h1}, 9^{h2}]$$

$$[2^{h1}, 10^{h3}, 6^{h2}, 1^{h1}, 18^{h3}, 1^{h1}, 9^{h2}]$$

**b. Division / Aggregation**

$$[2^{h1}, 10^{h3}, 6^{h2}, 18^{h3}, 1^{h1}, 1^{h1}, 9^{h2}]$$

$$[2^{h1}, 10^{h3}, 6^{h2}, 8^{h3}, 10^{h3}, 1^{h1}, 1^{h1}, 9^{h2}]$$

**c. Misalignment**

$$[2^{h1}, 10^{h3}, 6^{h2}, 18^{h3}, 1^{h1}, 1^{h1}, 9^{h2}]$$

$$[1^{h1}, 2^{h1}, 10^{h3}, 6^{h2}, 18^{h3}, 1^{h1}, 1^{h1}, 9^{h2}]$$

**Figure 4: H123 Fingerprint Characterizes.** Here for the sake of simplicity we have compressed the http version information into a one-dimensional representation.

set, retaining only samples whose resource quantities fall within 20% of the target sample's total. This optimization reduces unnecessary distance calculations, resulting in an approximately 85% reduction in comparisons during our experiments.

*3.2.2 Distance Metrics: Wasserstein and LCSS.* Accurate classification in WF attacks requires distance metrics that capture the unique characteristics of encrypted web traffic. Our analysis of H123 fingerprints identified three primary forms of intra-class variability: sequence permutation, division/aggregation, and temporal misalignment (as shown in Figure 4). These variations, though subtle, are intrinsic to fingerprint data and must be accounted for in our classification method. To address these issues, we designed a composite distance metric using **Wasserstein distance** [77] and **LCSS (Longest Common Similar Subsequence) distance**, tailored specifically to handle these types of variability.

**Wasserstein Distance** (also known as Earth Mover's Distance) measures the difference between two probability distributions. It calculates the minimum "work" required to transform one distribution into another, considering the spatial relationship between elements. This makes it particularly suitable for addressing small shifts in sequence elements or aggregation/division patterns in fingerprint data. The distance is calculated using the following formula:

$$W(P, Q) = \inf_{\gamma \in \Gamma(P,Q)} \int |x - y| d\gamma(x, y) \tag{2}$$

where $P$ and $Q$ are probability distributions, and $\gamma$ represents a transport plan.

**LCSS Distance** measures local similarity by identifying common subsequences between two sequences. It captures variations in temporal sequences, such as the misalignment of elements, which are common in H123 fingerprints. We introduce a relaxed similarity threshold, allowing for slight deviations between elements to account for these shifts. The similarity criterion, denoted as $\epsilon$, ensures robustness against small misalignments between subsequences. The adjusted LCSS distance improves sensitivity to local patterns while handling sequence-level discrepancies effectively.

The combination of these two metrics, weighted appropriately, provides a tailored measure for comparing website fingerprints, improving classification accuracy and robustness. Detailed pseudocode for the implementation of these distance measures is provided in Appendix E.

*3.2.3 Classification Strategy.* In a closed-world scenario, where all categories are known, we adopt the classical Nearest Neighbor (NN) approach to classify a test sample based on the closest reference sample. This method provides high classification accuracy within the range of known categories.

In contrast, the open-world scenario introduces the challenge of unknown categories. To address this, we combine the K-Nearest Neighbors (KNN) algorithm with the Gini coefficient [12] to assess whether a test sample belongs to any reference category. Specifically, we calculate the distances to the $2N$ nearest reference samples for the test sample, where $N$ represents the number of reference samples per category.

The Gini coefficient, which measures inequality, helps us assess the distribution of these distances. If the test sample belongs to a known category, we expect the $2N$ nearest samples to include $N$ samples with much smaller distances, representing the same category, and $N$ samples with larger distances, representing other categories. This leads to an unequal distribution of distances and a lower Gini coefficient. Conversely, if the test sample does not belong to any known category, the distances will be more uniformly distributed, leading to a higher Gini coefficient. By setting a threshold on the Gini coefficient, we can reliably distinguish between samples that belong to the monitored set and those that do not.

## 4 EXPERIMENTAL EVALUATION

### 4.1 Datasets

In the absence of an appropriate public dataset for HTTPS WF research, we created a comprehensive dataset that encompasses 12 distinct scenarios and includes over 220,000 website visits. This robust dataset serves as a platform for evaluating the efficacy of both the H&W method and state-of-the-art attacks. Details of the dataset are presented in Table 1.

**Closed-World (CW) dataset.** As in most related work [33, 49, 70], we first construct attack scenarios for **popular** websites based on the most recent Tranco top-site ranking list [56]. We focus on and collect data from the top 1600 websites, which, to our knowledge, constitutes the closed-world dataset with the largest number of categories involved. In addition, we consider more realistic scenarios of interest to attackers, including a **random** selection of 320 websites from the top 10k websites of the Tranco list; and a scenario that simulates network **censorship** by conducting a WF attack on 320 websites from the Citizenlab list [43], which are known to be susceptible to such scrutiny.

**Bandwidth dataset.** For our experimental setup across network conditions, we have designed a collection process from the top (Tranco list) 200 websites under three scenarios informed by recent network measurement studies [9, 57, 73]: **unconstrained**, **starlink-like**, and **slow**. Subsequently, we conduct cross-validation to assess the robustness of our method within these varied conditions. For details on the selected network conditions and implementation, refer to Appendix F.

**Browser dataset.** As in related work [35, 68], we selected the two most prevalent cross-platform browsers [74], Google Chrome and Mozilla Firefox, to construct our dataset. Given that Microsoft Edge utilizes the same rendering engine as Chrome, it was deemed non-distinctive for our purposes and thus excluded. To capture a

Table 1: **Overview of datasets.** For OW datasets, one sample per website; for other datasets, 40 samples per website.

| Task | Scenario | # Websites | # Samples | Task | Scenario | # Websites | # Samples |
|---|---|---|---|---|---|---|---|
| **CW** | popular | 1600 | 64,000 | **Browser** | chrome | 200 | 8,000 |
| | random | 320 | 12,800 | | chrome-legacy | 200 | 8,000 |
| | censorship | 320 | 12,800 | | firefox | 200 | 8,000 |
| **Bandwidth** | unconstrained | 200 | 8,000 | **Time-Drift** | 18-day | 200 | 8,000 |
| | starlink-like | 200 | 8,000 | | 30-day | 200 | 8,000 |
| | slow | 200 | 8,000 | **OW** | open world | 80,000 | 80,000 |

broader range of user update practices, we also included a **legacy Chrome** (v104) along with the latest **Chrome** (v126) and **Firefox** (v129). Across different scenarios, we collected traffic data from the top (Tranco list) 200 websites.

**Time-Drift dataset.** During the data collection process, we purposefully introduced time intervals, which naturally led to the above three datasets exhibiting concept drift. Specifically, we initiated the collection of the **Bandwidth** dataset on the 18th day and the **Browser** dataset on the 30th day following the completion of the **CW** dataset. To construct a temporal sequence dataset, we selected the **CW-popular** dataset (0 day) as a temporal reference and combined it with the **Bandwidth-unconstrained** dataset (18 days later) and the **Browser-chrome** dataset (30 days later). Consistency in the collection environment across these datasets ensures the comparability of our temporal analysis.

**Open-World (OW) dataset and Supplementary data.** We selected the first 80,000 unique websites from the Tranco list that were not previously collected to serve as the **open-world** scenario. Additionally, we have supplementary data consisting of the top 400 websites, each with 40 samples, collected half a year ago. This additional data only serves two specific purposes: (1) independent hyperparameter tuning for the H&W method, and (2) acting as essential pre-training data for Triplet Fingerprinting [71].

During the data collection process, we followed the methodology established in prior work [5, 49, 70], excluding websites that failed to load, displayed CAPTCHA, had no meaningful content, or involved fewer than 50 packets. These sites are generally not of interest to attackers since they lack practical relevance for users [49]. In line with previous research [66, 68], we employed an automated approach using *Selenium* (v4.15.2) to control Chrome browser (v126) (except for the browser dataset) and visit the homepage of each website. This method produces more realistic traffic compared to tools like *wget* or *curl*. The network traffic generated during the process was captured using *tcpdump*.

Following the recommendation in [39], we used a round-robin fashion during data collection to ensure more realistic and representative traffic. We deployed 10 Amazon cloud servers running Ubuntu 24.04 to execute the data collection in parallel batches. For the open-world scenario, each website was accessed once, whereas for other scenarios, we collected 50 samples per website and retained 40 valid samples. After each round of data collection, we conducted a separate DNS resolution for all web resources to build IP-WF [33].

### 4.2 Experimental Setup

*4.2.1 Optimization of Hyperparameters.* We independently optimized hyperparameters for H&W using **supplementary data**,

Table 2: **Closed-world: Accuracy (%) of WF attacks across different datasets in a few-shot scenario.** We bolded results over 95% as sufficiently reliable for attacks and marked the best results in each scenario.

| Attacks | N=1 BST[1]=20 hours | | | N=2 BST=40 hours | | | N=3 BST=60 hours | | | N=10 BST=8 days | | | N=30 BST=25 days | | | Time Overhead[2] | |
|---|---|---|---|---|---|---|---|---|---|---|---|---|---|---|---|---|---|
| | pop. | rand. | cens. | pop. | rand. | cens. | pop. | rand. | cens. | pop. | rand. | cens. | pop. | rand. | cens. | train↓ | pred.↑ |
| k-FP [29] | 40.1 ± 0.6 | 50.3 ± 0.5 | 58.3 ± 0.9 | 60.2 ± 0.1 | 70.6 ± 0.4 | 76.9 ± 1.0 | 69.5 ± 0.2 | 80.3 ± 0.7 | 84.0 ± 0.5 | 84.2 ± 0.1 | 90.8 ± 0.4 | 92.8 ± 0.2 | 90.6 ± 0.3 | 94.9 ± 0.6 | 95.9 ± 0.3 | 10.9s | >5k |
| DF [70] | 16.7 ± 2.2 | 38.6 ± 1.2 | 29.0 ± 1.4 | 39.7 ± 1.9 | 53.5 ± 1.1 | 49.5 ± 0.7 | 58.0 ± 1.1 | 82.9 ± 0.4 | 62.1 ± 0.4 | 91.9 ± 0.3 | 96.9 ± 0.1 | 93.1 ± 0.5 | 98.0 ± 0.2 | 99.0 ± 0.2 | 98.6 ± 0.4 | 254.4s | 3,765 |
| TF [71] | 57.3 ± 0.4 | 69.7 ± 0.9 | 66.9 ± 0.7 | 67.9 ± 0.3 | 79.3 ± 0.8 | 74.3 ± 0.5 | 72.6 ± 0.4 | 83.3 ± 0.5 | 78.8 ± 0.5 | 79.7 ± 0.2 | 88.8 ± 0.4 | 83.3 ± 0.3 | 81.8 ± 0.3 | 90.9 ± 0.3 | 84.8 ± 0.3 | 205.9s | 873 |
| RF [65] | 10.6 ± 1.1 | 40.2 ± 0.9 | 33.2 ± 1.9 | 37.0 ± 1.0 | 56.9 ± 1.2 | 52.3 ± 1.4 | 46.1 ± 1.1 | 78.1 ± 0.4 | 74.0 ± 1.0 | 78.3 ± 0.5 | 91.1 ± 0.6 | 87.0 ± 0.8 | 89.5 ± 0.9 | 97.8 ± 0.4 | 96.0 ± 0.5 | 26.5s | >5k |
| k-FP+ | 64.1 ± 1.0 | 69.4 ± 1.2 | 76.5 ± 1.3 | 87.1 ± 0.3 | 90.8 ± 0.3 | 92.6 ± 0.3 | 93.0 ± 0.1 | 95.2 ± 0.5 | 95.8 ± 0.4 | 98.0 ± 0.1 | 98.3 ± 0.2 | 98.8 ± 0.2 | 99.1 ± 0.1 | 99.5 ± 0.1 | 99.4 ± 0.1 | 16.3s | >5k |
| DF+ | 15.6 ± 1.5 | 46.9 ± 2.1 | 38.7 ± 1.1 | 44.8 ± 1.2 | 56.6 ± 1.5 | 56.4 ± 1.2 | 63.6 ± 1.2 | 83.5 ± 0.5 | 68.8 ± 1.0 | 94.3 ± 0.3 | 97.6 ± 0.1 | 96.3 ± 0.4 | 98.5 ± 0.1 | 99.5 ± 0.1 | 99.4 ± 0.1 | 277.5s | 4,298 |
| IP-WF (primary) [33] | 52.4 ± 0.2 | 60.3 ± 0.2 | 74.7 ± 0.5 | 52.4 ± 0.3 | 62.7 ± 0.4 | 76.9 ± 0.5 | 53.2 ± 0.4 | 64.1 ± 0.4 | 77.4 ± 0.6 | 54.0 ± 0.3 | 66.4 ± 0.3 | 78.7 ± 0.3 | 54.9 ± 0.2 | 67.1 ± 0.2 | 79.2 ± 0.2 | - | >5k |
| IP-WF [33] | 65.2 ± 0.3 | 64.9 ± 0.2 | 81.4 ± 0.4 | 68.3 ± 0.6 | 67.8 ± 0.3 | 84.7 ± 0.4 | 69.4 ± 0.4 | 70.0 ± 0.5 | 85.6 ± 0.8 | 72.5 ± 0.2 | 73.8 ± 0.3 | 89.0 ± 0.3 | 73.4 ± 0.2 | 74.9 ± 0.1 | 90.2 ± 0.2 | 50.1s | 4,023 |
| IP-WF (ideal) | 79.5 ± 0.5 | 79.3 ± 0.6 | 85.2 ± 0.5 | 84.3 ± 1.0 | 84.9 ± 0.4 | 88.4 ± 0.5 | 87.0 ± 0.5 | 87.4 ± 0.5 | 89.5 ± 0.7 | 92.1 ± 0.3 | 93.8 ± 0.4 | 93.8 ± 0.7 | 93.9 ± 0.2 | 96.2 ± 0.2 | 95.0 ± 0.1 | 50.1s | 3,819 |
| HOLMES Cosine | 42.4 ± 1.6 | 48.9 ± 0.7 | 53.8 ± 1.8 | 52.2 ± 1.2 | 61.9 ± 1.1 | 62.2 ± 1.1 | 59.5 ± 0.8 | 69.2 ± 0.7 | 67.2 ± 0.8 | 75.0 ± 0.8 | 82.7 ± 0.4 | 82.1 ± 0.4 | 82.4 ± 0.6 | 91.2 ± 0.5 | 89.7 ± 0.5 | - | >5k |
| HOLMES Edit | 73.9 ± 0.8 | 83.3 ± 0.5 | 77.8 ± 0.4 | 82.6 ± 0.6 | 90.3 ± 0.7 | 85.5 ± 0.5 | 85.5 ± 0.6 | 93.3 ± 0.6 | 89.2 ± 0.7 | 93.3 ± 0.4 | 97.0 ± 0.3 | 94.2 ± 0.6 | 96.4 ± 0.4 | 98.6 ± 0.3 | 96.4 ± 0.2 | - | 1,091 |
| HOLMES Wasserstein | 80.1 ± 0.6 | 88.3 ± 0.6 | 82.4 ± 1.0 | 86.1 ± 0.7 | 92.9 ± 0.8 | 86.7 ± 0.6 | 88.3 ± 0.6 | 96.3 ± 0.5 | 89.1 ± 0.9 | 94.1 ± 0.4 | 98.1 ± 0.3 | 94.4 ± 0.6 | 96.6 ± 0.3 | 98.9 ± 0.2 | 95.5 ± 0.5 | - | >5k |
| HOLMES LCSS | 78.8 ± 1.1 | 87.6 ± 1.4 | 81.0 ± 1.2 | 87.9 ± 1.0 | 91.9 ± 0.7 | 88.7 ± 0.7 | 89.3 ± 0.6 | 95.2 ± 0.5 | 89.9 ± 0.8 | 95.1 ± 0.4 | 97.2 ± 0.5 | 95.1 ± 0.8 | 97.8 ± 0.4 | 98.7 ± 0.4 | 96.2 ± 0.6 | - | 2,819 |
| HOLMES WATSON | 85.3 ± 0.8 | 90.4 ± 1.0 | 87.3 ± 0.8 | 91.9 ± 0.6 | 95.8 ± 1.0 | 92.9 ± 0.7 | 94.4 ± 0.7 | 96.4 ± 0.5 | 94.6 ± 0.4 | 96.7 ± 0.3 | 98.6 ± 0.3 | 96.4 ± 0.3 | 98.7 ± 0.3 | 99.5 ± 0.2 | 97.6 ± 0.2 | - | 2,068 |

[1] Bootstrap time is the total time required to initiate a comprehensive WF attack, including data collection and model training.
[2] Time overhead when validated on popular datasets with N=3. We record the training time and prediction throughput (websites/second).

ensuring no involvement of test sets from later validation experiments to avoid data leakage. We ultimately selected the maximum length of the H123 fingerprint ($L$) as 50, set the similarity threshold for the LCSS distance ($\epsilon$) to 0.24, and assigned equal weights of 0.5 for both distance measures.

*4.2.2 WF attacks for comparison.* To make a comprehensive comparison, we selected 5 state-of-the-art WF attacks: k-FP [29], DF [70], TF [71], RF [65], and IP-WF [33]. Each was implemented using the authors' released code, and fine-tuned for fairness in our experiments. We also introduced several optimizations: enabling packet length features in k-FP for better performance in HTTPS scenarios (k-FP+), mapping packet length information onto trace sequences to enhance DF (DF+), and constructing an ideal IP-WF scenario by utilizing IP addresses from browser logs rather than traffic flows to eliminate interference, thereby estimating the upper accuracy limit of this attack. All WF attacks were executed on a server running Ubuntu 23.10, equipped with an NVIDIA A100 80GB GPU, Intel Xeon 2.9GHz CPU, and 128GB RAM. To avoid random errors, we applied k-fold cross-validation to all attacks.

*4.2.3 Metrics.* We adopt the evaluation metrics from [70]. In closed-world tasks **Accuracy** is used to assess the performance of WF attacks in multi-class classification of monitored websites. In open-world tasks, while WF attacks still involve multi-class classification, we focus more on their ability to distinguish monitored websites from unmonitored ones. Thus, we treat it as a binary classification problem and use **Precision**, **Recall**, and **F1-score** as evaluation metrics. Compared to TPR and FPR, these metrics are more robust in handling class imbalance, as they provide a more comprehensive assessment of both positive and negative class performance.

## 4.3 Closed-world Evaluation

We first evaluate H&W in a closed-world scenario, assuming the clients visit only a set of monitored sites of interest to the attacker. The evaluation results are presented in Table 2.

**Experiment 1: WF attacks with different data distributions.** Traditional WF attacks require a large, regularly updated dataset, leading to excessive *bootstrap time* that renders the attack impractical for adversaries [71]. We evaluated the performance of various methods under different training sample sizes, with N = {1, 2, 3, 10,

30}, where the first four settings represent typical few-shot learning tasks. Results show that H&W achieves an attack accuracy (*average across different monitored sets*) of 87.7% with N = 1 . This implies that using H&W, an attacker only needs to visit each monitored site once to launch an effective WF attack, significantly outperforming the best existing IP-WF method, which achieves 70.5%. As N increases, the accuracy of all WF attacks improves. With N = 30 (bootstrap time exceeding 25 days), k-FP+ achieves the highest accuracy of 99.3%. Although H&W was not designed for this scenario, it still achieves a top-tier performance of 98.6%, while the widely discussed IP-WF under HTTPS scenarios delivers a mediocre performance, achieving only 79.5%.

**Experiment 2: WF attacks with different monitored sets.** We evaluated the performance of various attacks across different monitored website sets to comprehensively assess their applicability in more realistic and diverse scenarios. H&W consistently performed well, achieving, e.g., 95.8% accuracy in the random set with 320 websites using only two training samples. There is a positive correlation between the number of websites in the monitored set and attack difficulty, as most attacks performed worse on the popular set compared to the random and censorship sets. In comparisons across sets with the same number of websites, H&W, DF, TF, and RF slightly outperformed in the random set, while k-FP performed better in the censorship set. IP-WF showed a notable advantage in the censorship set, likely due to the more distinct IP addresses of the websites, with less IP co-location observed.

**Experiment 3: Time overhead analysis.** The training time and prediction throughput of an attack are crucial factors that influence its practicality. Frequent changes in website resources necessitate periodic data updates and retraining, and long training times can compromise the attack's timeliness. We conducted experiments on a high-performance computing platform, recording both training times and prediction throughput (websites/second) for various attacks. Notably, H&W, a lightweight, distance-based lazy learning method, requires no parameterized model training, eliminating training time overhead and allowing flexibility for periodic updates or incremental learning. In contrast, other methods require extensive training (e.g., TF requires pre-training, and IP-WF needs entropy fingerprint training), with deep learning-based approaches typically being the most time-consuming. In the prediction phase,

we evaluated the throughput of each method, which better reflects their performance in real-world scenarios. Across all website finger-

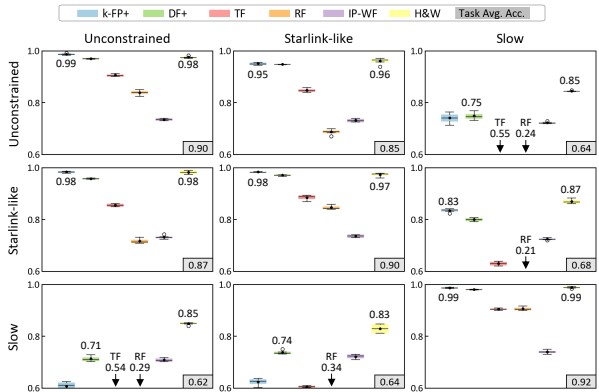

**Figure 5: Comparison of attack robustness across network conditions with $N$ = 10 in the Bandwidth dataset.**

printing attack methods, prediction throughput reached acceptable levels. H&W, for example, can handle at least 2,000 website visits per second, demonstrating its efficiency in high-traffic situations.

**Experiment 4: H&W with different distance metrics.** The results in table 2 show that Wasserstein distance and LCSS significantly outperform cosine distance and edit distance in distinguishing H123 fingerprints. By combining these two distances, WATSON achieves an average accuracy improvement of 3%, demonstrating their complementary nature in handling fingerprint features and enhancing overall performance.

## 4.4 Robustness Evaluation

Evaluating the robustness of attack methods in realistic settings is essential. As in prior work, we focus on cross-network conditions scenario [5, 65, 87] and concept drift scenario [5, 21, 33, 68].

**Experiment 1: WF attacks under cross-network conditions.** We performed cross-training and validation of various attacks on three typical network condition datasets: unconstrained, starlink-like, and slow. The results on the diagonal of Figure 5 show the performance of baseline attacks without cross-training, with $k$-FP+ and H&W achieving the best results. The off-diagonal results indicate that all attack methods experience varying degrees of performance degradation under cross-network conditions, with bandwidth changes (unconstrained-slow) having a significantly greater impact than packet loss changes (unconstrained-starlink). Notably, H&W performs better than existing methods under cross-network conditions, thanks to the application-layer resource features of H123 being independent of network conditions, which enhances its robustness. Additionally, $k$-FP+ and RF exhibit the most severe accuracy decline due to their features being strongly correlated with loading time, while IP-WF demonstrates the highest stability under cross-network conditions but with mediocre attack performance.

**Experiment 2: WF attacks under concept drift.** When a website's contents (such as images or ads) are updated, its traffic representation changes accordingly. This results in growing discrepancies between training data and actual attack data over time, a phenomenon known as concept drift [27]. Figure 6(b) illustrates

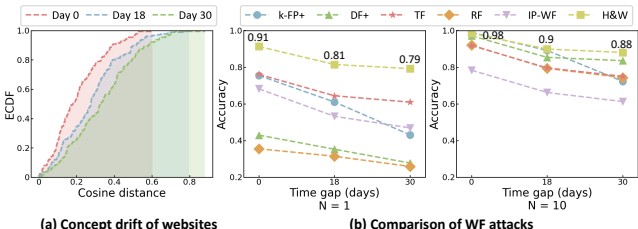

**Figure 6: Evaluation of concept drift in the Time dataset.** (a) Cosine distance measures concept drift's impact on traffic features, shown by the empirical CDF of average distances for the same site. (b) Shows the robustness and stability of different attacks under concept drift.

how the attack performance of the initial model varies after 18 and 30 days. The figure shows that as the time interval increases, concept drift intensifies, which directly leads to a decrease in attack performance across all methods. Furthermore, increasing the number of training samples (from N=1 to N=10) does not alleviate the effects of concept drift. Notably, with only 1 training sample and a 30-day interval, H&W still achieves nearly 80% attack accuracy. For N=10, although $k$-FP+ and H&W both achieve over 98% accuracy without concept drift, $k$-FP+'s accuracy drops to 72% after 30 days of concept drift, while H&W maintains 88% accuracy.

## 4.5 Open-world Evaluation

The open-world scenario offers a more realistic assessment of WF attacks by accounting for the vast, unexhausted range of websites users might visit. We simulate a more practical setting by using the **CW-censorship** dataset as the monitored set to reflect the potential of real-world network censors, and we consider $N$ = {1, 2, 3, 10} as training samples, with the rest used for testing. Initially, we evaluate various attacks using a random sample of 10k websites from the **OW** dataset as the unmonitored test set. We then further assess the impact of a larger open-world set on H&W performance.

**Experiment 1: WF attacks in the open-world scenario.** As shown in Table 3, in the open-world scenario, network censors using H&W can achieve over 85% micro F1 with just one training sample, significantly outperforming $k$-FP+ and DF+ in few-shot settings. While IP-WF excels in distinguishing between monitored and unmonitored sets, its performance declines when identifying specific websites. With increased training costs, H&W achieves 92.1% binary F1 and 94.2% micro F1 with N=10, slightly below the performance of $k$-FP+ and DF+.

**Experiment 2: H&W deep dive in the open-world scenario.** As with previous works [49, 70, 71], we analyze H&W's performance in larger open-world scenarios. We assert that increasing the number of open-world websites does not affect the training data but instead expands the number of negative samples in the test set. As the number of negative samples grows, more are incorrectly classified as positive, leading to a decrease in precision, though recall remains unaffected. Figure 7(a) shows the relationship between H&W's attack performance and the size of the open world for $N$ = {1, 10}. The P-R curve, representing attack performance, gradually declines as the number of negative samples increases.

Table 3: **Open-world: Binary and micro F1 (%) of WF attacks.**

| Att. | N = 1 binary[1] | N = 1 micro[2] | N = 2 binary | N = 2 micro | N = 3 binary | N = 3 micro | N = 10 binary | N = 10 micro |
|---|---|---|---|---|---|---|---|---|
| k-FP+ | 79.3 ± 0.8 | 77.9 ± 1.0 | 81.5 ± 0.4 | 87.1 ± 0.4 | 85.1 ± 0.5 | 90.4 ± 0.4 | 92.8 ± 0.3 | 95.8 ± 0.2 |
| DF+ | 46.0 ± 1.1 | 43.1 ± 1.1 | 65.0 ± 0.5 | 60.7 ± 0.8 | 81.4 ± 0.5 | 75.3 ± 0.6 | 93.9 ± 0.3 | 95.1 ± 0.3 |
| IP-WF | 88.1 ± 0.5 | 84.7 ± 0.4 | 89.5 ± 0.5 | 87.1 ± 0.4 | 90.1 ± 0.1 | 87.9 ± 0.3 | 91.2 ± 0.1 | 90.1 ± 0.2 |
| H&W | 84.7 ± 0.6 | 85.9 ± 0.6 | 88.2 ± 0.7 | 90.5 ± 0.7 | 90.2 ± 0.6 | 92.4 ± 0.5 | 92.1 ± 0.5 | 94.2 ± 0.4 |

[1] measures the attack's ability to distinguish monitored vs. unmonitored websites.
[2] measures detailed classification performance with website-specific identification.

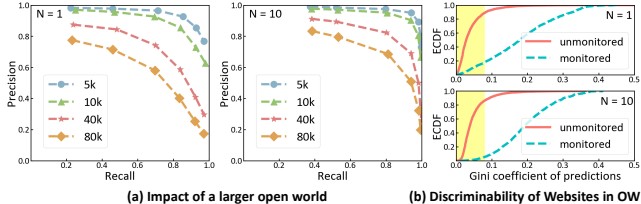

(a) Impact of a larger open world    (b) Discriminability of Websites in OW

**Figure 7: In-depth analysis of H&W in open world.** (a) Analyze the impact of unmonitored website quantity on H&W attack performance. (b) Examine the distinguishability between monitored and unmonitored sets in H&W and optimal threshold selection.

With 80,000 unmonitored websites, H&W still demonstrates reasonable performance under $N = 10$, achieving a precision of 70.2% and recall of 81.0%. Figure 7(b), on the other hand, shows the cumulative distribution curves of Gini coefficient values for samples processed by H&W in both monitored and unmonitored sets. The clear distinction between the two curves demonstrates H&W's strong discriminative ability.

## 4.6 Case Study: Applicability of H&W

H&W, as an attack method utilizing application-layer fingerprint features, presents an interesting and meaningful area of study regarding its applicability under various factors. We primarily explore the effects of different website characteristics, browser types, and defense strategies on H&W performance. To ensure generalizability, we conducted experiments on the **CW-random** and **Browser** datasets with $N = \{1, 10\}$.

We first analyzed target websites, collecting data on resource quantity, connection quantity, and HTTP version density (measured by the Shannon entropy [64] of the version sequence). The cumulative distribution of these metrics is shown in Figure 8. We constructed value intervals based on quartiles and conducted independent training and testing of H&W within these intervals. The results in Table 4 indicate that websites with resource and connection quantities in the (Q2, Q3] range are more vulnerable to the attack. Additionally, HTTP version density positively correlates with attack performance, meaning websites using multiple HTTP versions are more susceptible to H&W than those using only one. As websites adopt higher HTTP versions or as new versions emerge, version parallelism will intensify, making H&W increasingly threatening.

We validated H&W across three major browsers, showing strong performance on all. While it performed slightly worse on Firefox than Chrome, it still achieved over 98% accuracy with $N = 10$. We also evaluated the impact of padding-based defense strategies, finding that known defenses reduced attack accuracy by about 3%,

Table 4: **Case study: Impact of website characteristics.**

| Main Factors | | [0, Q1] | (Q1, Q2] | (Q2, Q3] | (Q3, max] |
|---|---|---|---|---|---|
| **Resource Quantity** | N = 1 | 83.8 ± 1.5 | 87.5 ± 0.6 | 91.1 ± 0.9 | 86.8 ± 1.7 |
| | N = 10 | 95.5 ± 1.1 | 98.6 ± 0.3 | 98.7 ± 0.3 | 98.1 ± 0.8 |
| **Connection Quantity** | N = 1 | 83.2 ± 1.0 | 87.8 ± 1.7 | 90.7 ± 1.3 | 88.2 ± 1.6 |
| | N = 10 | 94.9 ± 0.8 | 98.0 ± 0.8 | 99.1 ± 0.4 | 98.5 ± 0.3 |
| **HTTP Version Diversity** | N = 1 | 83.7 ± 1.4 | 88.8 ± 1.1 | 89.3 ± 1.4 | 90.6 ± 1.3 |
| | N = 10 | 96.2 ± 0.3 | 98.2 ± 0.5 | 98.3 ± 0.3 | 99.2 ± 0.2 |

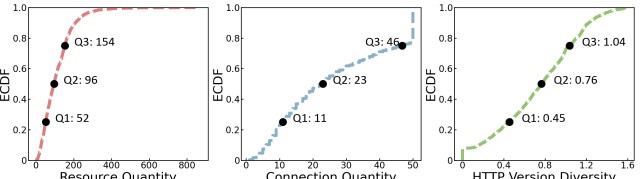

**Figure 8: Distribution of main factors in H123 fingerprints.** The distribution of resource and connection quantities is affected by the maximum fingerprint length ($L = 50$). HTTP version density is represented by the Shannon entropy of the version sequence.

maintaining a practical 95.5%. For detailed experimental setup and analysis, refer to Appendix G.

The above study demonstrates that H&W is not only lightweight and robust at the method level but also applicable across different server- and client-side environments. It is important to highlight that existing typical defense strategies are insufficient to mitigate H&W attacks, and with the growth of version parallelism, the threat will increase.

## 5 DISCUSSION AND CONCLUSION

This study utilized the phenomenon of HTTP version parallelism in HTTPS traffic to propose a novel fingerprinting method based on H123 fingerprints. We believe this approach can extend to other layers with *protocol version parallelism*, such as in TLS (e.g., TLS 1.2 and TLS 1.3) and the coexistence of IPv4 and IPv6. Combining multiple layers of protocol information could reveal more insights, further exacerbating privacy risks in encrypted traffic.

In our implementation of H123 fingerprints, we assumed equal importance for each fingerprint part. However, inspired by recent research on early detection [20, 90], we suggest assigning more weight to the earlier parts—given their stability and reliability—to enhance accuracy. Additionally, leveraging the characteristics of H123 fingerprints summarized in Figure 4 for data augmentation [5, 86] could further strengthen attack performance. This weighted approach and data augmentation represent promising directions for future work.

In conclusion, this paper introduced a method for HTTPS website fingerprinting using HTTP version parallelism, proving highly effective in small-sample scenarios and achieving robust performance with just one sample per monitored website. The experimental results demonstrated resilience across various network conditions and an ability to handle concept drift. As version parallelism grows with the adoption of different protocol versions, the privacy risks identified are likely to become more significant, highlighting the need for further research to mitigate these emerging threats.

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

## A  Related work

The foundational premise of WF asserts that unique traffic characteristics can distinguish websites, even when the traffic is encrypted or obscured. This concept emerged in the 1990s [79], and was first prominently featured in academic discourse by Andrew Hintz, who analyzed the number and sizes of web resources to conduct fingerprinting [31]. Concurrent studies also leveraged network resource metrics for differentiation [14, 75].

The advent of application-layer pipelining and multiplexing, alongside advancements in privacy technologies like the Tor network, rendered early fingerprinting techniques less effective, redirecting focus to transport-layer packet features. Herrmann et al. [30] employed frequency analysis of packet sizes in conjunction with Bayesian classifiers to execute fingerprinting within ssh tunnel. Subsequent research utilized edit distance on packet size sequences for website identification [11, 47]. Wang et al. combined a manual feature set of 4226 features with k-nearest neighbor classifiers for WF attacks [80], while Hayes and Danezis [29] proposed $k$-FP method, which used a refined set of 150 features with random forest models for the same proposal. The introduction of deep learning techniques to WF by [1] has marked a significant evolution in the field, with subsequent research leveraging a variety of advanced deep learning models. These include convolutional neural networks [7, 63, 70], triplet networks [71], and generative adversarial networks [51], establishing a new research trajectory and representing the state-of-the-art in performance.

Building on the extensive evolution of WF techniques over the past two decades, targeted research on refining fingerprinting features has emerged as a critical focus within the domain. Panchenko et al. [52] introduced the CUMUL fingerprint, which leverages cumulative representations of packet sizes. Rahman et al. [58] developed the tik-tok attack, significantly utilizing the temporal characteristics of website packets. Shen et al. [65] proposed a robust

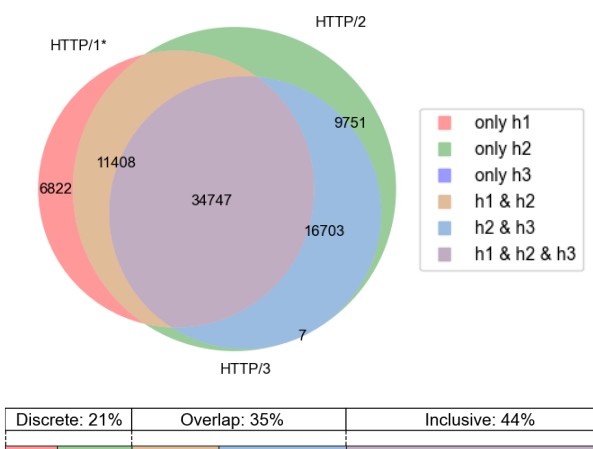

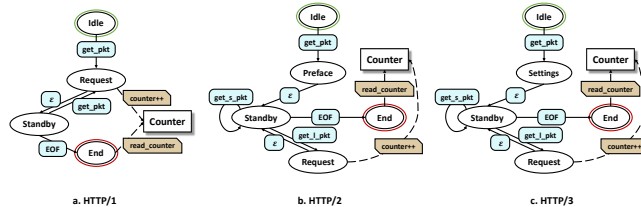

**Figure 10: Protocol State Machines Designed for Different HTTP Versions.** In the diagram, ellipses represent system states, rounded rectangles indicate state transition conditions, and rectangles with cut corners denote functions bound to a specific state.

Table 5: **Explanation of Protocol State Machine Elements**

| Identifier | Type | Description |
|---|---|---|
| Idle | state | Initial state of the protocol state machine |
| Request | state | Client requests a resource |
| Standby | state | Waiting for event trigger standby |
| Preface | state | Client sends preface packet |
| Settings | state | Client sends settings packet |
| End | state | Final state of the protocol state machine |
| get_pkt | transition | Retrieve any packet |
| get_s_pkt | transition | Retrieve short packet |
| get_l_pkt | transition | Retrieve long packet |
| EOF | transition | Process to end of file |
| $\epsilon$ | transition | Automatic state transition |
| Counter | variable | Resource quantity counter |
| counter++ | function | Increment operation on counter |
| read_counter | function | Read counter value |

Among these, over half of the websites show a relatively severe phenomenon: every version of the HTTP currently in use coexists among all their network resources. Additionally, we found that websites utilizing the HTTP/3 protocol almost invariably exhibit version parallelism, which can be attributed to the fact that new technologies are typically deployed incrementally. Our findings regarding the use of the HTTP/3 protocol diverge from other authoritative results [78]. This discrepancy arises because those studies focus on the homepage resource's support for the protocol, whereas our discussion encompasses the application of the protocol across all network resources of the website.

## C Inferring resource quantity using protocol state machines

Our protocol state machines keep meticulous track of the state of encrypted traffic, as shown in Figure 10. The state machines start processing packets from the client in the *idle* state and recognize resource requests based on the transition conditions defined in Table 5. When a packet meets these conditions and triggers a resource *request* state, the corresponding counter is incremented. This approach focuses only on the number of resources transmitted in the data stream and is not affected by encryption masking content information.

**Figure 9: HTTP Version Parallelism Measurement.** Venn diagram showing the percentage of parallelism of various http versions for top 100k sites. The discrete, overlap, and inclusive in the bar chart represent that these websites use only one http version, use two http versions, and use three http versions in their web resources, respectively.

fingerprinting feature based on time-window analysis. Beyond the traditional focus on transport layer information for WF, Siby et al. [69] introduced WF using side-channel encrypted DNS traffic; Nguyen et al. [33] constructed fingerprints combining IP addresses with browser rendering characteristics for large-scale WF. Li et al. [45] evaluated the website fingerprints themselves, proposing WeFDE method using information quantification theory to reasonably assess the information leakage of fingerprints, avoiding the potential biases of model selection and accuracy judgment.

The research most aligned with the methodologies in this paper involves inferring protocol semantics from encrypted traffic [3, 25, 42, 55, 84, 88], utilizing attributes not concealed by encryption.

## B Measurement of HTTP Version Parallelism across Top 100k Websites

In our study, we conducted extensive measurements of HTTP version parallelism on the real internet to demonstrate the widespread occurrence of this phenomenon. Specifically, we automated access to the top 100k websites from the Tranco top-site ranking list. By retrieving browser log information, we assessed whether the access attempts were successful and recorded the HTTP version information used for each network resource within a single visit. Our definition of HTTP version parallelism is as follows: if the set of HTTP versions used across all network resources within a website contains only one element, we refer to this as "discrete"; if the set contains two elements, we call it "overlap"; and if network resources use HTTP/1.1, HTTP/2, and HTTP/3 concurrently, we describe this situation as "conclusive".

The measurement results, as shown in Figure 9, indicate that nearly 80% of the websites exhibit HTTP version parallelism.

It is worth noting that the packets generated by the application layer control features introduced by HTTP/2 and HTTP/3 are independent of the number of network resources. We blocked these control packets using two main methods: (1) adding an initial step that does not inflate the counter to offset the effect of the *setting* packet, and (2) using packet size thresholds that filtered out control packets during resource transfers (in particular, packets below 40 bytes in HTTP/2 and packets below 120 bytes in HTTP/3).

## D  Examples of H123 fingerprints

```
                    [  1,  0,  0,  0,  0,  0] - h1
    google.com:     [  0,  1,  0,  1,  2,  1] - h2
          L=6       [  0,  0, 20,  0,  0,  0] - h3

                    [  0,  0,  0,  0,  0] - h1
   facebook.com:    [  3, 10,  0,  0,  0] - h2
          L=5       [  0,  0,  3,  9, 13] - h3

                    [  0,  0,  0,  4,  0, ...,   0,  0,  0] - h1
    youtube.com:    [  7,  0,  2,  0,  2, ...,   0,  0,  1] - h2
         L=20       [  0, 17,  0,  0,  0, ...,   3, 14,  0] - h3

                    [  0,  0,  0,  0,  0,  0] - h1
   instagram.com:   [  4,  0, 17,  0,  0,  1] - h2
          L=6       [  0, 26,  0,  1,  2,  0] - h3

                    [  0,  0,  0,  0,  0] - h1
   cloudflare.com:  [  2,  0,  1,  1,  0] - h2
          L=5       [  0, 22,  0,  0, 40] - h3

                    [  0,  0,  0,  6,  3, ...,   1,  0,  0] - h1
     fastly.net:    [  3,  0,  3,  0,  0, ...,   0,  1,  2] - h2
         L=52       [  0, 34,  0,  0,  0, ...,   0,  0,  0] - h3

                    [  0,  0,  0,  0,  0,  2] - h1
      bing.com:     [  3,  3,  0,  0,  1,  0] - h2
          L=6       [  0,  0, 82,  3,  0,  0] - h3

                    [  1,  0,  0,  0,  5, ...,   0,  5,  0] - h1
    spotify.com:    [  0,  3,  0,  2,  0, ...,   0,  0,  6] - h2
         L=23       [  0,  0,  2,  0,  0, ..., 115,  0,  0] - h3

                    [  0,  0,  0,  0,  0, ...,   0,  0,  0] - h1
   wordpress.com:   [  2,  2,  4,  4,  0, ...,   1,  1,  1] - h2
         L=15       [  0,  0,  0,  0,  2, ...,   0,  0,  0] - h3

                    [  1,  0,  0,  0,  0, ...,   0,  0,  0] - h1
      zoom.us:      [  0,  3,  6,  5,  2, ...,   1,  1,  3] - h2
         L=20       [  0,  0,  0,  0,  0, ...,   0,  0,  0] - h3
```

**Figure 11: H123 Fingerprint examples.** Some of the fingerprints in the figure are not shown completely due to length reasons, with "L" indicating the length of the fingerprints.

Figure 11 presents real examples of H123 fingerprints for the 10 most frequently visited websites. These fingerprints combine the sequence of HTTP versions and the sequence of resource quantity generated during a website visit, resulting in a final fingerprint form of 3×L dimensions (where rows represent different HTTP versions, and columns represent each network flow). These examples of H123 fingerprints demonstrate two characteristics: (1) H123 fingerprints are concise and direct, stemming from their direct representation of the characteristics at the website's network resource layer; (2) H123 fingerprints are distinguishable. The quantity of network resources and the distribution of HTTP versions vary across different websites, creating a vast fingerprint space.

## E  WATSON Distance-metric Pseudocode

Wasserstein distance, also known as Earth Mover's Distance (EMD), is a method used in mathematics and data science to measure the difference between two probability distributions and is widely applied in various scenarios. We interpret the H123 fingerprint as a discrete distribution and calculate the one-dimensional Wasserstein distance for resource sequences of different HTTP versions, averaging the three results in the end. The one-dimensional Wasserstein distance for discrete distributions, as required in our case, has been proven to be equivalent to the cumulative distribution function (CDF) distance between two sequences [59], which greatly enhances computational efficiency and theoretical simplicity. Furthermore, based on preliminary experimental results, we decided to sort the sequence data numerically before computing the Wasserstein distance. The pseudocode for calculating the Wasserstein distance is shown in Algorithm 1

---

**Algorithm 1** Wasserstein Distance

---

**Require:** Two H123 fingerprints $F_1$ and $F_2$
**Ensure:** Normalized Wasserstein distance
1: $W \leftarrow 0$ {Sum of Wasserstein distances}
2: $T \leftarrow 0$ {Total sum of $F1$ elements for normalization}
3: **for** $i = 0$ to 2 **do**
4:     $C_1 \leftarrow$ ComputeCumulativeDistribution(Sort($F_1[i]$))
5:     $C_2 \leftarrow$ ComputeCumulativeDistribution(Sort($F_2[i]$))
6:     $W \leftarrow W + \sum_j |C_1[j] - C_2[j]|$ {Accumulate the absolute differences}
7:     $T \leftarrow T + \sum_j F_1[i][j]$ {Accumulate the total sum of $F_1$ elements}
8: **end for**
9: $W \leftarrow \frac{W}{T}$ {Normalize the Wasserstein distance}
10: **return** $W$

---

**Algorithm 2** LCSS Distance

---

**Require:** Two H123 fingerprints $F_1$ and $F_2$, similarity threshold $\epsilon$
**Ensure:** Normalized LCSS distance
1: Initialize LCS matrix $dp$ with size $(len(F_1) + 1) \times (len(F_2) + 1)$
2: **for** $i = 1$ to $len(F_1)$ **do**
3:     **for** $j = 1$ to $len(F_2)$ **do**
4:         $threshold \leftarrow \epsilon \cdot \max(F_1[i-1], F_2[j-1])$ {Calculate similarity threshold}
5:         **if** $|F_1[i-1] - F_2[j-1]| \leq threshold$ **then**
6:             $dp[i][j] \leftarrow dp[i-1][j-1] + 1$
7:         **else**
8:             $dp[i][j] \leftarrow \max(dp[i-1][j], dp[i][j-1])$
9:         **end if**
10:     **end for**
11: **end for**
12: Compute LCSS by tracing back through $dp$ matrix
13: **return** $1 - \frac{\text{sum of LCSS}}{\text{sum of } F_1}$

---

The Longest Common Subsequence (LCS) algorithm is a method used to find the longest shared subsequence between two sequences.

Table 6: **Detail of network conditions.**

| Scenario | Upload Bandwidth | Download Bandwidth | Network Latency | Packet Loss Rate |
|---|---|---|---|---|
| **Unconstrained** | >500mbps | >500mbps | <10ms | <1% |
| **Starlink-like** | 8mbps | 70mbps | 50ms | 6% |
| **Slow** | 300kbps | 800kbps | 350ms | 3% |

It has broad applications in fields such as text comparison and data mining. In our application scenario, sequence elements (the number of resources) may exhibit minor variations due to various reasons, making the traditional LCS algorithm less suitable, as it requires exact matching of sequence elements. To address this issue, we propose the Longest Common Similar Subsequence (LCSS) distance, which introduces a similarity threshold, allowing for certain variations in the elements of the sequence. This distance focuses more on assessing the overall similarity of non-sparse sequences, and we choose to use network resource sequence fingerprints (which do not explicitly label HTTP versions) for measurement. The pseudocode for calculating LCSS distance is shown in Algorithm 2.

## F  Bandwidth dataset

Cross-network condition scenarios (where there are significant differences between the network environments during training and testing data collection) are considered a major challenge in making WF attacks practical [5, 15, 87]. Since it is impossible to cover all possible network conditions, we designed three network scenarios with typical differences and used them for cross-validation of the attack method (training the model in one network condition and testing it in others).

We primarily considered the following four network performance metrics: Upload Bandwidth, Download Bandwidth, Network Latency and Packet Loss Rate. As shown in Table 6, the first scenario is **unconstrained**, representing a good network environment typical for most users. This setup is based on network measurement results from Cloudflare Radar [57]. The second scenario is **starlink-like**, simulating users accessing websites via Starlink's commercial satellite internet service. According to the measurement results in [9], Starlink is characterized by relatively slow speeds and higher packet loss rates. The final scenario is **slow**, which simulates conditions in countries with poor network infrastructure or in situations of sustained slow network speeds due to congestion, characterized by low bandwidth and high latency.

During data collection phase, to accurately simulate these network conditions, we referred to the Chrome Developer documentation [22] and used Chrome's **network throttling** feature to create custom network conditions.

## G  Case study: browsers and defense strategies

In this section, we will provide a detailed overview of H&W's applicability across different browsers and the impact of existing defense mechanisms on its attack performance.

### G.1  Attack performance on different browsers

We conducted cross-training and testing of H&W on the **Browser** dataset, with results presented in Table 7. Training and testing on

Table 7: **Evaluation results on different browsers.**

| Scenario | Individual | | Cross-version | | Cross-browser | |
|---|---|---|---|---|---|---|
| | N = 1 | N = 10 | N = 1 | N = 10 | N = 1 | N = 10 |
| **Chrome** | 88.4 ± 0.6 | 98.6 ± 0.3 | 62.2 ± 1.2 | 73.3 ± 1.2 | 4.9 ± 0.9 | 6.8 ± 0.5 |
| **Chrome-legacy** | 87.5 ± 0.9 | 98.2 ± 0.3 | 58.9 ± 1.1 | 74.6 ± 1.2 | - | - |
| **Firefox** | 83.6 ± 1.8 | 98.0 ± 0.5 | - | - | 4.2 ± 0.9 | 6.3 ± 0.7 |

the same data distribution (within the same browser) yielded reasonable results across different browser scenarios; for instance, in Chrome with $N = 1$, the attack accuracy was 88.4%. When $N$ increased to 10, the attack accuracy exceeded 98% across all browsers. We observed a slight decrease in accuracy on Firefox, likely due to the greater diversity of control frames triggered during resource loading, which complicates resource inference.

Initially, we expected H&W to demonstrate strong cross-browser capability since it extracts application layer resource features. However, the actual results showed high sensitivity to changes in data distribution when switching browsers. In cross-browser version experiments, we noted about a 20% drop in accuracy, and H&W became ineffective in cross-browser tests. This is attributed to significant differences in how Chrome and Firefox handle website resources, leading to noticeable discrepancies in the resource sequences extracted by HOLMES for the same website.

We must emphasize that cross-browser applicability remains an open problem in website fingerprinting attacks. Similar to the results in [68], we conducted the same cross-browser validation for $k$-FP+ and obtained comparable outcomes, with an accuracy of 6.5% when $N = 10$. To address the challenges of cross-browser scenarios, we suggest incorporating samples from multiple browsers during the training phase or using JA4 fingerprinting [46] to identify the browser before conducting the attack. However, these considerations are beyond the scope of this paper and will be left for future work.

Table 8: **Evaluation results under padding-based defense.**

| Scenario | k-FP+ | | DF+ | | H&W | |
|---|---|---|---|---|---|---|
| | N = 1 | N = 10 | N = 1 | N = 10 | N = 1 | N = 10 |
| No defense | 69.4 ± 1.2 | 98.3 ± 0.2 | 46.9 ± 2.1 | 97.6 ± 0.1 | 90.4 ± 1.0 | 98.6 ± 0.3 |
| Defense with prior know. | 50.3 ± 0.5 | 90.8 ± 0.4 | 38.6 ± 1.2 | 96.9 ± 0.1 | 81.0 ± 1.2 | 95.5 ± 0.7 |
| Defense without prior know. | 3.5 ± 1.0 | 6.4 ± 0.6 | 2.1 ± 1.1 | 5.9 ± 0.6 | 5.7 ± 0.9 | 6.9 ± 0.9 |

### G.2  Attack performance under defense strategies

In existing research on anonymous networks, various advanced WF defenses have been proposed to counter WF attacks. These strategies typically introduce perturbations to standard traffic characteristics by adding redundant packets [28, 40], delaying transmissions [10, 82], or splitting application flows [19]. However, the use of these defenses in anonymous networks relies on bidirectional cooperation within the system, meaning that the perturbations added by the Tor client must be removed at the relay nodes so that the traffic reaching the actual server is free of these disturbances. In contrast, there is no such communication mechanism or protocol in standard HTTPS that enables coordinated defense between the client and server to implement these advanced traffic defenses.

Packet padding is a simple yet effective method of traffic obfuscation [41]. In the context of standard HTTPS, some studies

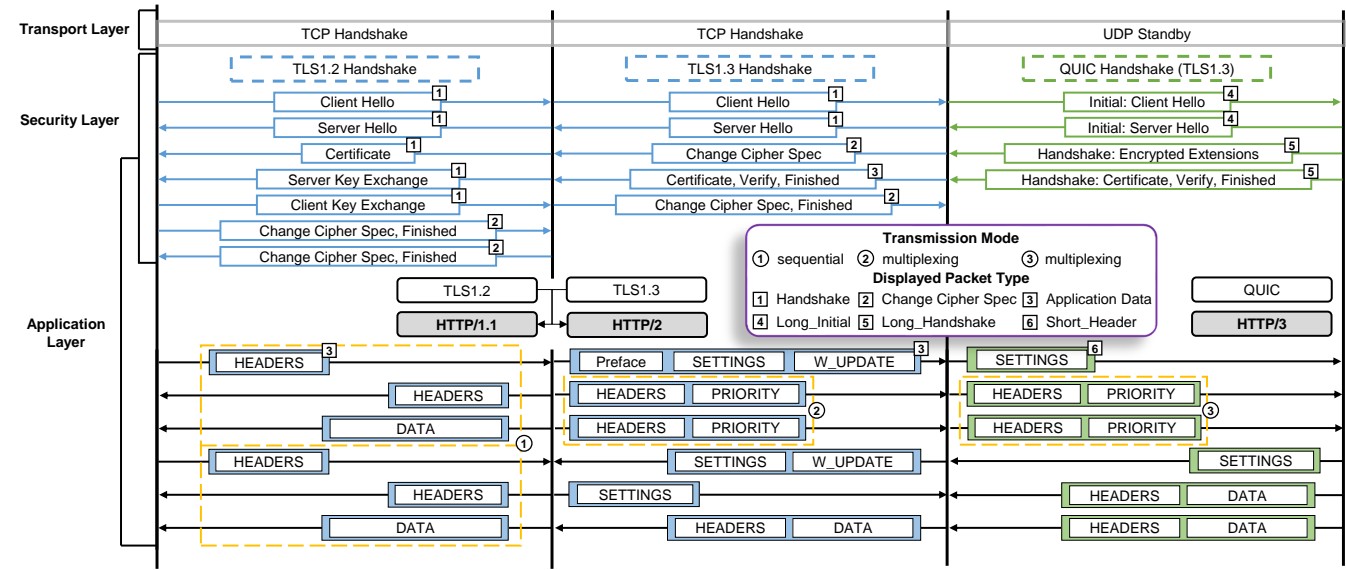

**Figure 12: Network Protocol Stack in HTTP Version Parallelism Scenario.** HTTP/1.1 and HTTP/2 protocols can use TLS 1.2 and TLS 1.3, respectively, as their security layers; HTTP/3 can only use QUIC as its security layer.

have explored using the padding mechanism provided by QUIC at the transport layer to introduce server-agnostic defense perturbations [68, 72]. Although this technique has not yet been adopted by mainstream browsers, we simulate this feasible defense strategy and evaluate its impact on WF attacks. Specifically, we use the **CW-random** dataset, padding each packet to its maximum length to construct a defense dataset. Existing work often assumes that attackers have prior knowledge of the defense strategies being implemented [65, 70], meaning they can train models on the defense dataset. We also consider a more challenging scenario where the attacker is unaware of the defense strategy.

We evaluated the performance of $k$-FP+, DF+, and H&W under the defense mechanism. First, we trained and tested on the dataset without packet padding as a performance baseline. Next, we assessed the performance of various methods when the attackers

had prior knowledge of the defense strategy. As shown in Table 8, packet-padding defenses did not effectively mitigate website fingerprinting attacks under adversarial conditions. For instance, when $N = 10$, the accuracy of H&W remained at 95.5%, only about 3% lower compared to the scenario without defense.

In the experiments where attackers were unaware of the defense implementation, most website fingerprinting methods became nearly ineffective, losing practical utility. This is because the packet-padded test data introduced significant distribution differences compared to the original training data. However, it is important to emphasize that users cannot rely on attackers' ignorance of the defense strategy. Instead, efforts should focus on developing more robust defense mechanisms.

