# OpenReview forum: "HOLMES & WATSON: A Robust and Lightweight HTTPS Website Fingerprinting through HTTP Version Parallelism"
_ACM.org/TheWebConf/2025/Conference — WWW 2025 Oral_

### Official Review · Reviewer_VpBG · 2024-11-17

**Novelty:** 6
**Technical Quality:** 5

**Review:**

## Summary

The paper introduce a HTTPS website fingerprinting method called HOMES & WATSON. It makes use of the fact that different HTTP versions are offered by different web services. The methodology was evaluated with a large dataset containing 80,000 popular websites from the Tranco list crawled in different scenarios. The evaluation showed that the fingerprinting method is very reliable to be used for tracking website visits in encrypted HTTP traffic.

## Pros

- Well thought attack with high impact
- Evaluated with studies reflecting real-world scenarios
- Paper and methodology easy to follow for most parts
- Open Data set and source code released to community

## Cons

- Study justification misses some important details
- No Limitations stated
- Results could be discussed and justified better
- Minor details missing, tables could be improved



## Explanation

Thank you for submitting this work to WWW '25! The proposed fingerprinting methods is realistic and can be used by actors that restrict access to the free Internet or that like to identify user that access certain websites on the Internet. Therefore the submission is of high relevance and impact. Also, the conducted experiments are realistic and resemble real-world scenarios where this tracking method could be used. Furthermore, I like that the paper is easy to follow, and that most figures explain the attack and methodology in an understandable way. The fact, that the data set and algorithm will be made available to the public, is appreciated.

Nevertheless, there are some changes that should be addressed. I hope that the feedback will help the authors to increase the quality of the paper and make it publishable.

### Study Setup Justification

It seems to me that mobile users were not considered in the study setup, although they are pretty common. Also, the Safari browser is also not considered in the study setup although it has the second biggest market share in online browsers in the desktop scenario (~13%). Why did you exclude them from the study setup?

The study set up misses some details. When were the data sets collected? What is the exact Tranco data set that you used (Tranco offers permalinks to the different data sets)? How did you collect the browser data set (as it did not use Selenium)? What were the server locations and Internet service providers that you used to crawl the different websites for the tracking mechanism? Why did you only use Chrome legacy and not Firefox legacy? What was the setup of the used high-performance cluster? How was the currency calculated? How did you fine-tune the algorithms for fairness in the experiments?

### Lack of Limitations

Every scientific work has its limitations. However, the paper does not critically discuss any. I would expect that the authors critically discuss the limitations of their work. One of them could be for example the location of the AWS servers. For instance: Does the geolocation or the browser vendor have an impact on the overall fingerprinting per website or the general website behavior?

### Table Design

The results tables, especially table 2 is very confusing for me as a reader. It's took multiple attempts to me to understand what it is about and what results which part _might_ relate to (although not always 100% sure). What experiment (1,2,3) is located where in the table? Also, there are a lot of numbers in the table. To reduce the mental overload, I would suggest to include the full table in the appendix and a shortened version in the main text. The shortened version could then just show a "-" instead of the real number for all values that are lower than 95% (as it seems that all values lower than 95% are irrelevant anyways).

### Result Discussion and Justification

There are a lot of results that are presented in the paper, which is great. Nevertheless, I would've expected a larger discussion of all of these results. Especially it would be interesting to know what the results could mean for me as a potential reader (e.g., a developer, decision maker, browser vendor, or online service).

Experiment 3 in section 4.3 states that H&W "can handle at least 2000 website visits per second, demonstrating its efficiency in high traffic situations". Since the authors were only able to achieve this using a high-performance cluster, I would doubt that this is a realistic assumption as 2000 website visits per second happen at Internet service providers, that could be wiretapped by attackers, in a millisecond.

Some statements in the results are a bit vague or missing justification. For instance, "there is a positive correlation between the number of websites in the monitored set and attack difficulty". By what criteria did you calculated correlation and what are the p ans R^2 values for that? Also, when discussing that one method performs better than another, I would expect a significance testing.

"The results on the diagonal of Figure 5 show the performance of baseline attacks without cross-training, with k-FP+ and H&W achieving the best results.": To me, it seems like this is not always the case. When taking a look at Figure 5, k-FP+ underperforms in Slow/Unconstrained and Slow/Starlink-like.

### Minor

- Figure 1: I would love to see the citations to the original sources here, so that the readers can compare the made statements with the literature.
- "Juarez et al. [15]" does not reference Juarez at al. but Cherubin et al.
- Threat Model: Virtual private networks (VPNs) are commonly used to circumvent networks censoring. Does your threat model also includes this scenario?
- Section 4.1 inventions three different data sets called unconstraint, starlink-like, and slow. Although the full explanation is in Appendix F, it could be helpful for readers to add a short sentence describing these data sets in the main text.
- It would have been helpful for me to add to the description of Table 3 that these results were made using the CW-censorship data set.
- Section 4.6 introduced "padding-based defense strategies" without explaining them in more detail.

**Questions:**

- It seems to me that mobile users were not considered in the study setup, although they are pretty common. Also, the Safari browser is also not considered in the study setup although it has the second biggest market share in online browsers in the desktop scenario (~13%). Why did you exclude them from the study setup?

- The study set up misses some details. When were the data sets collected? What is the exact Tranco data set that you used (Tranco offers permalinks to the different data sets)? How did you collect the browser data set (as it did not use Selenium)? What were the server locations and Internet service providers that you used to crawl the different websites for the tracking mechanism? Why did you only use Chrome legacy and not Firefox legacy? What was the setup of the used high-performance cluster?

- Can the authors state the Limitations section that they would include in the next version of the paper?

**Reviewer Confidence:**

3: The reviewer is confident but not certain that the evaluation is correct

**Scope:**

4: The work is relevant to the Web and to the track, and is of broad interest to the community

---

### Official Review · Reviewer_sQ4q · 2024-11-18

**Novelty:** 6
**Technical Quality:** 6

**Review:**

### Summary
The paper proposes a new website fingerprinting approach (called HOLMES) that exploits application-layer features, namely HTTP version parallelism.
The approach works as follows:
- On the network layer an adversary is able to observe the HTTP version of each request.
- Many websites have unique orders of sequences of HTTP version requests. For example, the first request is HTTP/1.1, then there are two HTTP/3 requests, then three HTTP/2 requests, and then another HTTP/3 request.
- Using this sequence of HTTP version information (called H123 fingerprint), websites can be fingerprinted.

The authors then create a classifier (called WATSON) that uses lazy learning and the Wasserstein and LCSS distance metrics to classify whether a given sample belongs to any known website.
Lastly, the authors perform a comprehensive evaluation of their new classifier on a newly created dataset and compare it with various related approaches, showing that it has high accuracy and outperforms prior work.

### Evaluation
The paper proposes an interesting new way to perform website fingerprinting and shows that this new approach works better than existing techniques under various experimental settings. It remains a bit unclear how good the real world performance of the approach really is, but the results seem promising. Additionally, several ideas for future extensions and optimizations are mentioned.
In general, the paper is easy to follow and understand, however some addressable writing issues exist and some important details are only mentioned in the appendix.

### Pros
- Created new dataset for HTTPS WF research, containing 1.5TB of HTTPS traffic/220k samples for 80k websites (promised to publish)
- Novel fingerprinting idea: H123 fingerprints
- Lightweight classifier
- Comprehensive evaluation (various settings, robustness evaluation, performance evaluations, various other approaches) that shows superior results of this new approach
### Cons
- The classifier is not robust against browser version/type changes (only mentioned in the appendix!)
- Only landing pages/homepages were tested and no internal pages
- Some writing issues: e.g., the paper abuses the appendix, some figures are unused, no ethics section, no limitations section (see additional comments)

### Additional Comments

- It is great that some code is already available for reviewers and that the dataset will be published. Please consider publishing the dataset and code permanently (e.g., Zenodo) and not only on a institutional website that could turn of at anytime. Also it would be great if the benchmark suite with the implemented competing algorithms could be published as well.
- The sections about the uniqueness and stability of fingerprints (over time) is quite short (e.g., 4.4 Experiment 2). Also, I would expect a much higher degree of randomness in the fingerprints (e.g., due to ads). It would be great if you could extend the analysis here and show how stable the fingerprints are over time.
- The appendix seems abused and used for more or less essential parts of the paper:
    - The related work is in the appendix!
    - The explanations of the key distance metrics are in the appendix.
    - Examples of the fingerprints are in the appendix.
    - The important result that the approach is only usable for a fixed browser and version (changing the version results in 20% accuracy loss and changing the browser results in results similar to random guessing) is only mentioned in the appendix. In the main text this sounds quite different "We validated H&W across three major browsers, showing strong performance on all. [...] robust at the method level but also applicable across different server- and client-side environments". Please add the important fact that the good performance only applies when the attacker model is trained on the same browser that is used by the victims to the main text.
- Misleading presentation of the dataset with 12 distinct scenarios although only 10 datasets were collected. If I understood the text correctly, the time-drift 18-day scenario is the same as the bandwidth-unconstrained scenario and the time-drift 30-day scenario is the same as the browser-chrome scenario.
- It is a bit unclear why the hyperparameters were chosen (paper only says optimized using supplementary data) and what the benefits or issues with for example a higher max-length would be.
- Cross-Version/Browser comparison:
    - If the cross-version performance is quite bad, timedrift of the websites in scales >15 days could be irrelevant as all browsers auto-update and a new dataset for the new version would be required anyways?
    - How is the cross-browser performance between similar browsers (e.g., Chrome, Brave, Opera, Edge) that use the same engine? Are the differences (only) caused by the engine or also by the browser?
- Missing details:
     - The **K** for k-fold cross-validation is missing. Was the common value 10 used?
     - The sample size in the OW experiment is not stated. I assume it is the same as in the CW experiment and 50/50 samples were used?
 - Presentation:
     - In Figure 5 it is almost impossible to see the colors as they are very thin, please try to make the figure easier to digest.
    - Several figures are not really used in the text:
        - Some are not mentioned in the text at all (Figure 6(a))
        - Others are only shortly mentioned but not explained (Figure 1, 2, 3):
        - If you do not use the figures, consider removing them.
        - Otherwise, consider explaining the figures in more depth in the text and use them for your advantage. For example, Figure 1 could be used to explain the various WF approaches that exist.
    - A limitation section would be great.
    - An ethics section would be great, in particular as a dataset is open-sourced.
    - p.13 "cross-browser version experiments" reads confusing as I first read it as "(cross-browser) (version)" and not as "cross-(browser version)", maybe simply use "cross-version"?
- References and citations:
    - [37] links to the full archive of the mailinglist, it should point to a thread/mail which supports what it claims to support in the text.
    - p.10 "WF by [1]" should instead be "WF by Abe and Gota [1]"
    - The Tranco citation uses the wrong link (arxiv) and additionally misses the date and ID of the generated list that was used: See "referencing the list" on the Tranco website (https://tranco-list.eu/list/XJVLN/1000000)

**Questions:**

- Please provide more details on the real world feasibility of the attack. For example how unique are website patterns and how stable are they over time?
- Why is  H&W not compared to any other approaches in the larger open-world scenario?
- (Packed-Padding defenses) Could it be a suitable defense if it would be randomized across users as the attacker cannot train for all possible paddings?
- Further minor questions are in the "additional comments" section

**Reviewer Confidence:**

3: The reviewer is confident but not certain that the evaluation is correct

**Scope:**

3: The work is somewhat relevant to the Web and to the track, and is of narrow interest to a sub-community

---

### Official Review · Reviewer_vXBP · 2024-11-30

**Novelty:** 7
**Technical Quality:** 5

**Review:**

The authors investigate a novel website fingerprinting technique based on the co-existence of multiple HTTP versions in the wild. Each website requests a set of resources to servers configured using various versions of the HTTP protocol in unique, identifiable sequences. These fingerprintable patterns are compared via Wasserstein and (custom) LCSS distances to determine whether an observed sequence corresponds to a known website. Comprehensive experiments demonstrate that this technique generally outperforms existing solutions, while showing robust results in open and closed world scenarios, as well as over extended periods.

Although the paper contains numerous experiments, they are laid out relatively clearly, with an emphasis on comparison with the state-of-the-art. The technical solution is both novel and valuable to the research community. However, it would benefit from clearer contextualization of the threat model, a more robust hyperparameter tuning phase and a discussion of its various limitations

**Pros:**
- The fingerprinting technique presented in the paper is both novel and promising. While previous solutions focused on other network layers or features of the HTTP protocol, the use of HTTP versions for identification has not been explored. As the authors note, it should remain effective over time due to the co-existence of multiple protocol versions in parallel.
- The design itself is an interesting engineering feat. Lazy learning enables quick website classification with decent accuracy in a closed-world setting, requiring less training data than other existing solutions.
- The various datasets are comprehensive and aim to be realistic.
- Equally, the paper stands out for its numerous experiments, evaluating various fingerprinting use cases. Although information-dense, the results section efficiently conveys the main findings and compares them to the state-of-the-art. The authors' effort to update existing solutions for fair comparison is particularly appreciated.

**Cons:**
- Although quick website fingerprinting is technically interesting, it lacks motivation and clear context for the overall threat model. If an attacker can access ISP-level web traffic for a day, it seems reasonable to assume they could access similar logs over a month. As the paper notes, the lazy learning approach allows for an easily updated model, adapting to changes and new websites. However, this adaptability may be less relevant from a privacy standpoint.
- The hyperparameter tuning is performed on the top 400 websites from "half a year ago" to "avoid data leakage." Several other datasets also contain the top 400 websites (`CW`, `bandwidth`, `browser`, `time-drift`, and to a lesser extent, `OW` and `random`), which weakens the independence claim. While there is some concept drift over time, this alone may not be enough to ensure data leakage did not occur.
- The paper should more clearly discuss the approach’s limitations. First, although HTTP versions are incrementally deployed, concentrated platforms like Cloudflare and CDNs enable large-scale adoption in a few months `[1*]`, reducing fingerprint stability if a new version is released. Second, fingerprints will inevitably lose uniqueness over time as servers adopt the latest protocol version (until HTTP/4 is available). Third, the lack of cross-browser capability, discussed only in Appendix G, seems relevant and should be included in the main text. Finally, time-based features used in `[58]` could potentially complement this approach.
- Certain technical choices in the implementation or presentation should be further explained or justified:
    - The 20% band-pass optimization.
    - The prediction throughput (2,000 websites per second); it is described as acceptable but is one of the lowest compared to the state-of-the-art. For example, k-FP+ processes over 5,000 websites per second with similar accuracy (for N=3, Table 2). This should be further discussed and/or included in the limitations.
    - The selection of an 8-day dataset (N=10) in Experiment 1, Section 4.4, and why other options were not considered for the bandwidth dataset.

Below are suggestions for further improving the paper, in no particular order:
- Section 4.2.2 could benefit from an appendix containing a table summarizing the features and machine learning methods used by the studied state-of-the-art solutions.
- While padding and countermeasures are not the main focus of the paper, they could benefit from further explanation and study:
    - Clarify which part of the system padding affects. For models using packet length as a feature, this is straightforward. However, since H&W relies only on the inferred version of the HTTP protocol, it is not immediately clear how padding impacts the final predictions.
    - Padding up to the maximum payload represents a worst-case scenario from an attacker’s perspective, but it is unrealistic due to the additional overhead. The overhead should be measured, and alternative padding approaches could be explored (e.g., RFC 8467 `[2*]` or `[3*]`) .

`[1*]`: https://blog.cloudflare.com/cloudflare-view-http3-usage/
`[2*]`: https://datatracker.ietf.org/doc/rfc8467/
`[3*]`: Bushart, Jonas, and Christian Rossow. "Padding ain't enough: Assessing the privacy guarantees of encrypted {DNS}." _10th USENIX Workshop on Free and Open Communications on the Internet (FOCI 20)_. 2020.

**Questions:**

- How did you ensure the dataset used for hyperparameter tuning was not a subset of the training datasets?
- How is padding affecting H&W?

**Reviewer Confidence:**

3: The reviewer is confident but not certain that the evaluation is correct

**Scope:**

4: The work is relevant to the Web and to the track, and is of broad interest to the community

---

### Official Review · Reviewer_u11L · 2024-12-03

**Novelty:** 6
**Technical Quality:** 6

**Review:**

# HOLMES & WATSON: A Robust and Lightweight HTTPS Website Fingerprinting through HTTP Version Parallelism

## Summary

This paper presents HOLMES, a novel approach to Website Fingerprinting (WF) that improves the identification of websites from encrypted traffic, addressing the limitations of existing methods in real-world HTTPS conditions. Traditional WF techniques often struggle with network variability and concept drift, leading to performance degradation. HOLMES enhances robustness by exploiting HTTP version parallelism, extracting application-layer features that provide up to 4.28 bits of information—significantly outperforming previous methods in terms of stability and information gain. These features offer better resistance to network changes, improving the technique's ability to identify websites under dynamic conditions.

In addition to HOLMES, the paper introduces WATSON, a lightweight classification method based on lazy learning, which reduces the need for large-scale training datasets and computational resources. The combination of HOLMES' enhanced feature extraction and WATSON's efficient classification significantly improves the accuracy and efficiency of website fingerprinting. With an average accuracy of 87.7%, HOLMES and WATSON outperform state-of-the-art methods by over 15%, achieving this performance with only a single sample per website. This makes the proposed approach both highly efficient and robust, paving the way for more practical deployments of website fingerprinting in real-world environments.

* * *

## Pros

This paper tackles a critical challenge in the field of website fingerprinting (WF)—improving the robustness and efficiency of website identification using encrypted traffic. The authors introduce HOLMES, a novel approach that utilises HTTP version parallelism to extract enhanced application-layer features, which are demonstrated to surpass 98% of previously reported features in terms of stability across varying network conditions. The paper makes a valuable contribution by addressing both the network variability and concept drift issues that have historically limited the effectiveness of WF techniques in real-world HTTPS environments. Additionally, the introduction of WATSON, a lightweight classification method based on lazy learning, effectively reduces the reliance on large datasets, enhancing the practical deployment potential of the approach.

The experimental evaluation is one of the paper's key strengths, demonstrating a performance improvement over existing methods. Achieving an average accuracy of 87.7% with only a single sample per website is impressive, showcasing HOLMES and WATSON's ability to deliver both high robustness and efficiency. This is a noteworthy contribution, as the accuracy and sample size reduction mark clear advancements in WF research. The paper provides a comprehensive explanation of the methodology, ensuring transparency and reproducibility, which are essential for further academic investigation and practical deployment.

* * *

## Cons

While the paper makes a good contribution, there are a few areas where further elaboration would strengthen the work:

- Section 2.2, on threat modelling, is a reasonable assumption to have a passive eavesdropper; however, the authors do not discuss any insight into the attacker's computational capabilities. The WF can be used by state actors for censorship (which the authors discussed later on) but did not mention in their threat model.
- In section 3.2., the authors mention they opted to use K-NN with a Gini coefficient for efficiency. This is to ensure the samples are treated fairly and that the lazy learning approach manages to maintain a balanced distance across samples in binary classification. While this is reasonable, I wonder why the authors used this instead of other fair classification approaches such as disparate impact, which tries to balance out the confusion matrix across different groups.
- In section 4.2.2., the authors mentioned they fine-tuned the algorithms to their new dataset. While this is a correct approach, they do not discuss details of such fine-tuning. As I understand, the methods are adjusted to a protocol-level investigation different from the one in this paper. Thus, how do they ensure such adjustments are mitigated in their fair analysis?
- While focusing on HTTP versions is reasonable, there are more complications in real-world applications. The authors discussed briefly about TLS and IP versions. I expected a more comprehensive discussion on these matters, especially if combined with H123 fingerprints.
- Scalability and Generalisation: While the results are impressive, the scalability of HOLMES and WATSON to a larger set of websites or in real-time environments is not fully explored. A discussion of potential limitations or challenges related to scaling the approach and its generalisation to a broader range of use cases would add depth to the analysis. The idea of concept drift does not get the evaluations it deserves. I believe this is one of the contributions of the paper, so I expect to see a more robust explanation. This is important as advertisements across the web change rapidly.


### Suggestions for Improvement:

Include more technical details on the feature extraction, classification methods, and distance metrics to provide a clearer understanding of the inner workings of HOLMES and WATSON.

- Perform a detailed comparison with other state-of-the-art methods, providing a broader analysis of performance metrics, including precision, recall, and computational efficiency, across various conditions.

- Expand the discussion on the scalability and potential real-world deployment of HOLMES and WATSON, addressing possible challenges and suggesting future research directions.


* * *

## Conclusion

This paper makes a significant and timely contribution to the website fingerprinting domain, addressing key challenges related to robustness, efficiency, and dataset requirements. HOLMES and WATSON offer a promising solution that is both practical and effective, with the potential for real-world deployment in HTTPS traffic analysis. The methodology is innovative, and the experimental results are impressive, demonstrating a clear performance improvement over existing approaches. However, additional technical details and a deeper comparison with other methods would enhance the paper’s comprehensiveness.

**Questions:**

- Why Gini coefficient is a better option for fair lazy training?
- How do you ensure other WF methods generalise well with your dataset for a fair comparison? You mentioned fine-tuning but did not elaborate much.

**Reviewer Confidence:**

3: The reviewer is confident but not certain that the evaluation is correct

**Scope:**

4: The work is relevant to the Web and to the track, and is of broad interest to the community

---

### Official Review · Reviewer_98Pd · 2024-12-03

**Novelty:** 3
**Technical Quality:** 4

**Review:**

Summary
---------------
From the introduction:
"The increasing adoption of HTTPS has significantly improved user
privacy by encrypting communication content."
This paper seems to view that as undesirable and makes a concerted and decently successful effort to undo these gains.


Weaknesses
----------------
- Being the bad guys

  It's not entirely clear why this paper is aiming to be the bad guy,
  but the fabled quote from Jurassic Parc does come to mind.


Overall evaluation
----------------
The technical quality of the work seems decent enough, and I appreciate the paper's forthrightness in explaining their goal of undermining gains in user privacy achieved thanks to encryption. I'm just not really convinced this is a thing to strive for - and the paper does nothing to persuade me otherwise.

Other than that, this seems like yet another contribution to the field of website fingerprinting using ML. Decently enough executed, with a modest new spin on where to get the training data for the ML model from.

**Questions:**

-

**Ethics Review Description:**

This research seeks to undermine widespread gains in user privacy, without explaining why.

**Ethics Review Flag:**

Yes

**Reviewer Confidence:**

3: The reviewer is confident but not certain that the evaluation is correct

**Scope:**

3: The work is somewhat relevant to the Web and to the track, and is of narrow interest to a sub-community